# Direct programming of confined surface phonon polariton resonators with the plasmonic phase-change material In$_3$SbTe$_2$

Lukas Conrads [1,2] ✉, Luis Schüler [1,2], Konstantin G. Wirth[1], Matthias Wuttig [1] & Thomas Taubner [1] ✉

Tailoring light-matter interaction is essential to realize nanophotonic components. It can be achieved with surface phonon polaritons (SPhPs), an excitation of photons coupled with phonons of polar crystals, which also occur in 2d materials such as hexagonal boron nitride or anisotropic crystals. Ultra-confined resonances are observed by restricting the SPhPs to cavities. Phase-change materials (PCMs) enable non-volatile programming of these cavities based on a change in the refractive index. Recently, the plasmonic PCM In$_3$SbTe$_2$ (IST) was introduced which can be reversibly switched from an amorphous dielectric state to a crystalline metallic one in the entire infrared to realize numerous nanoantenna geometries. However, reconfiguring SPhP resonators to modify the confined polaritons modes remains elusive. Here, we demonstrate direct programming of confined SPhP resonators by phase-switching IST on top of a polar silicon carbide crystal and investigate the strongly confined resonance modes with scanning near-field optical microscopy. Reconfiguring the size of the resonators themselves result in enhanced mode confinements up to a value of $\lambda/35$. Finally, unconventional cavity shapes with complex field patterns are explored as well. This study is a first step towards rapid prototyping of reconfigurable SPhP resonators that can be easily transferred to hyperbolic and anisotropic 2d materials.

Tailoring the light-matter interaction at the nanoscale is one of the main goals of nanophotonics. The research area of plasmonics facilitates subdiffractional field confinement by exploiting metallic nanostructures. However, those structures suffer from extensive losses due to the very short lifetime of the excited surface waves called surface plasmon polaritons[1]. This obstacle can be solved by exploiting surface phonon polaritons (SPhPs) occurring in polar crystals, facilitating low-loss nanophotonics. The dielectric function of polar crystals offers a narrow region called Reststrahlenband between the longitudinal and transversal optical phonon frequency with a negative value of the real part and a small imaginary part of the dielectric function. SPhPs can be excited at an interface of a dielectric and the polar crystal and comprise coupled photons with collective oscillations of atomic cores. One of the most prominent polar crystals is silicon carbide (SiC) with the Reststrahlenband ranging from 797 – 973 cm$^{-1}$ [2–7]. A large variety of applications has been demonstrated in the past years, for example in thermal management[8,9], metamaterials[10,11], sensing[12,13], or non-linear optics[14].

Unfortunately, the functionality and design wavelength of those devices is fixed after fabrication and cannot easily be altered afterwards.

Dynamic functionalities can be achieved by employing different mechanisms[15] such as the photoexcitation of free charge carriers in SiC[16], gating graphene[17–19], intercalation[20,21] or twisting[22–25] layers of the

[1]Institute of Physics (IA), RWTH Aachen University, D-52056 Aachen, Germany. [2]These authors contributed equally: Lukas Conrads, Luis Schüler. ✉e-mail: conrads@physik.rwth-aachen.de; taubner@physik.rwth-aachen.de

polariton hosting material, or combining SPhPs with active materials. For example, the temperature-dependent insulator-to-metal transition of VO$_2$ enables tuning of the wavelength of phonon polaritons[26]. However, local modifications are not possible due to the volatile and temperature-induced phase transitions.

Phase-change materials (PCMs) are prime candidates for non-volatile tuning based on a change in the refractive index between two (meta-) stable phases, e.g., the amorphous phase and the crystalline one. Remarkably, both phases show a large contrast in the refractive index, caused by a change in chemical bonding. In the amorphous phase, the atoms are covalently bonded, but in the crystalline phase a new bonding type occurs which is called metavalent[27–31]. The reversible phase change can be induced by locally heating the PCM with electrical or optical pulses. Consequently, those materials have been utilized for reconfigurable polariton optical elements such as lenses[32] and for switchable SPhP resonators[33,34].

Recently, the plasmonic PCM In$_3$SbTe$_2$ (IST) was introduced[35], which can be reversibly switched from a dielectric amorphous state to a metallic crystalline one in the entire infrared range. In particular, the permittivity of crystalline IST is negative and follows a Drude-like behavior, as shown in Fig. 1A. In the crystalline phase, the electrons are strongly delocalized and enhance the conductivity of the material. The conductivity is about $10^4$ S/cm and therefore allows the classification as a bad metal. Previously, arbitrary nanoantenna shapes were fabricated via direct laser writing and even a large-area spatial emissivity shaping metasurface was demonstrated[36–41]. However, reconfiguring the resonators to modify the confined polaritons modes and investigating the coupling of metallic IST resonators to polaritons remains elusive.

Here, we demonstrate direct writing of confined SPhP resonators via laser irradiation with the plasmonic PCM IST (see Fig. 1B) on top of the polar crystal silicon carbide (SiC). Therefore, the PCM is locally heated with a focused laser above the glass transition temperature to crystallize the material. Crystallizing the amorphous IST (aIST) around the targeted resonator shape offers a significant degree of freedom for programming arbitrary shapes of amorphous IST within a crystalline surrounding. First, we investigate freely propagating SPhPs launched by an edge of crystalline IST, followed by confined SPhP modes of circular cavities with scattering-type scanning near-field optical microscopy (s-SNOM). Reducing the diameter of the fabricated cavities results in a spectral shift of the observed resonance modes. Afterwards, the field confinement is determined and stronger confinements for smaller cavities are obtained. Finally, we study arbitrary resonator shapes such as squares and triangles to highlight the vast flexibility of programming different shapes with the plasmonic PCM IST.

## Results

### Propagating surface phonon polaritons

First, we investigate freely propagating SPhPs launched by a laser-crystallized edge of crystalline IST with s-SNOM (c.f. Figure 2A). S-SNOM enables subwavelength resolution achieved by probing the sample with highly localized near-fields of a sharp tip. The near-field amplitude and phase are detected by demodulating the scattered electric field of the oscillating tip illuminated by infrared laser light. Therefore, the 35 nm thin amorphous IST layer on top of the polar crystal SiC is crystallized with precise laser pulses (see Methods). The propagating SPhPs are launched at the crystalline edge and propagate along the amorphous IST. Figure 2B displays the measured optical amplitude images $s_2$, normalized to crystalline IST, for frequencies varied from 880 to 930 cm$^{-1}$. Bright periodic fringes appear next to the crystalline IST, corresponding to SPhPs. Upon increasing the frequency, the spacing between these fringes increases. The polariton wavelength is determined by extracting line profiles along the measured s-SNOM images (see Fig. 2C) and applying a fit for a damped oscillating function with an additional linear background (see Supplementary Note 2 for more details and a zoom-in for better visibility of the fitted function onto the measured data. Moreover, the limited segment chosen for fitting at small frequencies is due to topographical artifacts not corresponding to polariton fringes. More details about this and an additional data set of measured propagating SPhPs can be found there). The measured propagation wavevectors from Fig. 2C are compared with theoretical calculations of the dispersion of SPhPs in Fig. 2D. The green curve represents the calculated slow-guided mode and the brown curve the fast-guided mode of the polaritons excited at the three-layer interface SiC/aIST/air. Both modes occur due to the chosen complex $k_{sp}$ ansatz (see Supplementary Note 3 for more information)[33]. Here, the calculated solution only takes three layers into account by neglecting the capping layer on top of the IST layer. The applied capping layer is taken into account by calculating the imaginary part of the reflection coefficient $r_p$ (color-coded map in Fig. 2D) with the transfer matrix method (see Supplementary Note 3). The maximum of the imaginary part of the reflection coefficient corresponds to the excitation of SPhPs and is close to the calculated slow mode of the polaritons. The experimentally extracted data (red dots with black error bars) match well with the simulated imaginary part of the reflection coefficient. The error bars are extracted from the applied fits of the

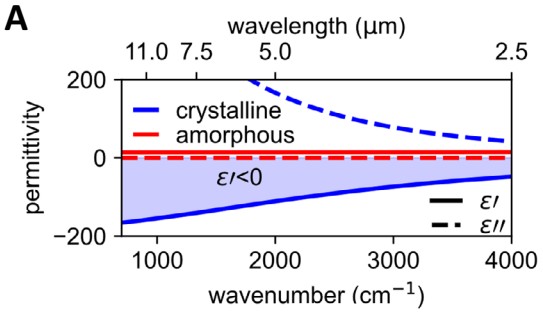

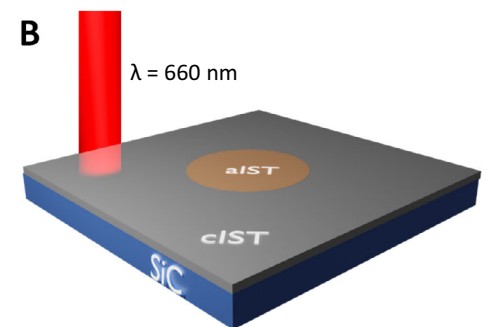

**Fig. 1 | General concept of programming resonators with IST. A** Dielectric function of amorphous and crystalline IST. Upon crystallization, the PCM changes from an amorphous dielectric to a crystalline metallic state. **B** The PCM is crystallized via laser irradiation to form amorphous cavities of IST on top of a SiC substrate.

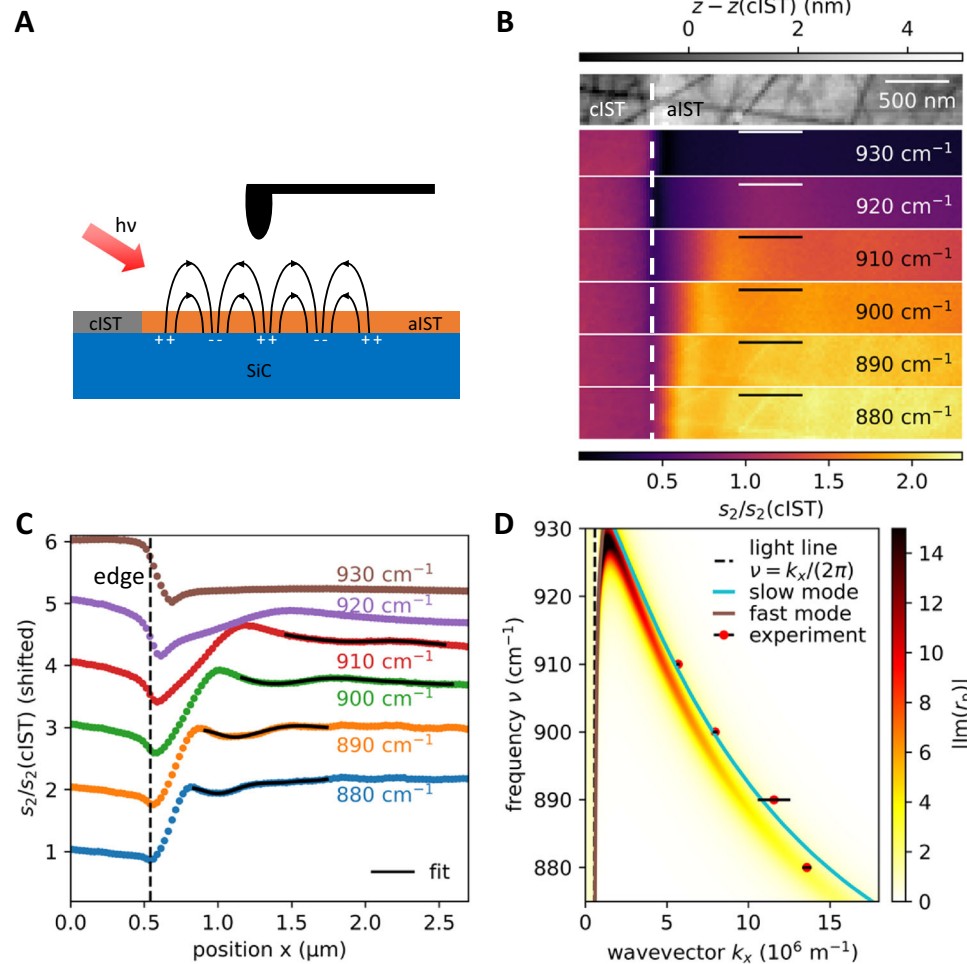

**Fig. 2 | Propagating SPhPs launched by a cIST edge. A** Surface phonon polaritons are launched at the laser crystallized edge and are detected with s-SNOM. **B** Optical near-field amplitude images $s_2$ and the corresponding AFM image (top) of the SPhPs for different excitation frequencies. The images are normalized to the signal of crystalline IST. **C** Averaged line profiles of the amplitude near-field images $s_2$ taken from **B**. The black line corresponds to a fitted function to determine the line profiles. The black line corresponds to a fitted function to determine the wavevector of the polaritons. **D** Dispersion plot of the SPhP on a SiC substrate covered with amorphous IST and capping. The green curve represents the expected slow mode of the polaritons for a layer stack without capping. A more adequate solution is given by the imaginary part of the reflection coefficient (color-coded) with capping considered. The experimentally obtained values are shown as red dots with black error bars.

line profiles. The measured propagation length of the SPhPs is in the range of $1\,\mu\text{m}$, allowing for programming confined resonators of similar size.

**Confined surface phonon polariton resonators**

Next, we optically write a circular SPhP resonator of amorphous IST on SiC with a diameter of $D = 3\,\mu\text{m}$ by crystallizing the material around the targeted structure (see Methods for more details). The measured s-SNOM near-field amplitude images with additional cross-section profiles and the topography can be seen in Fig. 3A. The excitation frequency is continuously decreased from $935\,\text{cm}^{-1}$ to $875\,\text{cm}^{-1}$. The near-field amplitude is normalized to the signal of the surrounding crystalline IST, which does not exhibit any spectral features. A pronounced and confined field enhancement appears at $920\,\text{cm}^{-1}$. Decreasing the excitation frequency leads to an evolution of the observed amplitude pattern inside the resonator. In particular, a ring mode structure is visible at $910\,\text{cm}^{-1}$. This pattern further transforms for an excitation frequency of $900\,\text{cm}^{-1}$ into a ring with enhanced fields in the center. For smaller frequencies, no mode structure can be observed.

Afterwards, we decreased the diameter of the cavity by applying more crystallizing laser pulses to investigate the effect of smaller cavities on the observed resonance modes (see Fig. 3B). The procedure

of creating those resonator shapes with sophisticated spatially overlapping laser pulses is explained in Supplementary Note 4. The corresponding s-SNOM amplitude images for the different cavity diameters with additional light microscope images allowing for a better distinction between the crystalline and amorphous phase can be found in Supplementary Note 5. Moreover, we performed numerical field simulations of the out-of-plane electric field component resembling the experimentally observed field patterns very well (see Supplementary Note 6).

The optical near-field amplitude in the center of the cavity for varied frequencies and different cavity diameters is shown in Fig. 3C. All curves display a maximum in the optical amplitude at the center which is shifted towards smaller frequencies for reduced cavity diameters. In addition, a cut-off is observed for wavenumbers larger than this peak. The quality factor $Q$ is determined by fitting a Lorentzian function and reveals values up to 50 (see Supplementary Note 7) in accordance with literature values of similar systems[6,34] and can be further improved by reducing the IST film thickness. Theoretically, the resonance frequency of such circular cavities can be calculated with $k_{\text{sp}}(\nu)D + \phi = 2x_n(J_m)$[42], where $k_{\text{sp}}(\nu)$ is the polariton wavevector, $D$ corresponds to the cavity diameter, and $\phi$ and $x_n(J_m)$ denote the reflection phase and the $n$th zero of the Bessel function $J_m$, respectively. We derived a reflection phase of $-\pi$ which confirms the

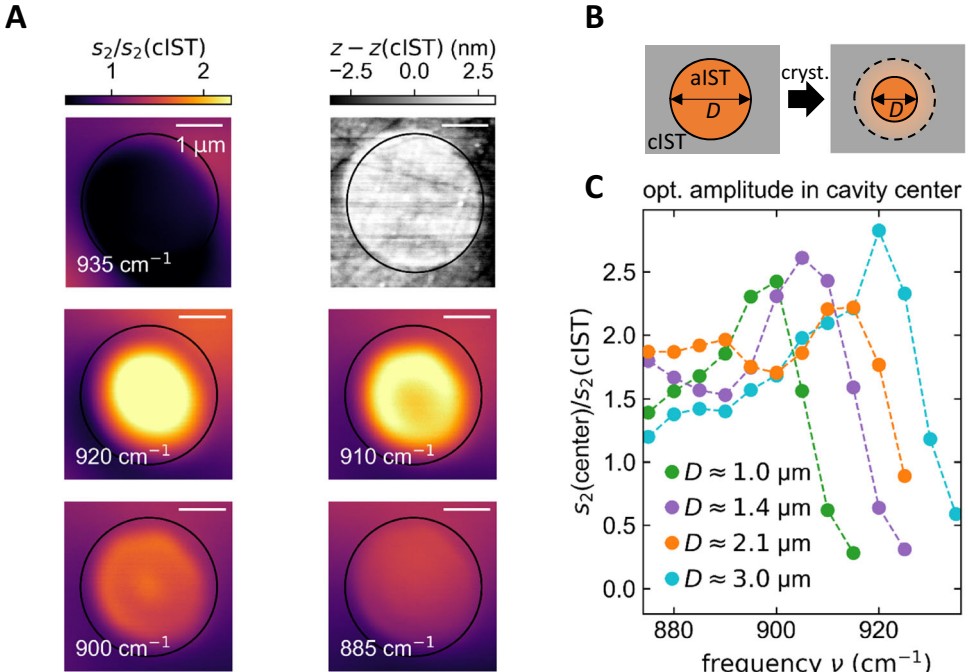

**Fig. 3 | Circular cavities with varying resonance mode profiles dependent on the cavity diameter. A** Normalized near-field amplitude images $s_2$ of a cavity with $D = 3\,\mu m$ for excitation frequencies from 875 cm$^{-1}$ to 935 cm$^{-1}$. Only a few representative s-SNOM images are displayed. The upper right image displays the topography of the amorphous cavity with a height of 2.3 nm. **B** Adding more

crystallizing pulses at the edge of the cavity results in reconfigured cavities with a smaller diameter $D$. **C** Measured normalized near-field amplitude in the cavity center for different diameters as a function of the excitation frequency. For smaller diameters, the peak shifts towards smaller frequencies.

expectations due to the metallic boundaries of the cavity (see Supplementary Note 8 for more details)[6,43].

With our presented technique of directly writing cavities featuring various resonance modes, programming of antenna arrays is possible as well. These arrays display clear resonances in the far-field spectra and are shown in Supplementary Note 9.

For further insights of the confined fields inside the cavities, we evaluated the lateral extension of the measured near-field amplitude at the corresponding resonance frequencies for different diameters in Fig. 4. While for all different cavities, a pronounced maximum is observed at their resonance frequency, the lateral extension varies. Reducing the cavity diameter results in more confined fields. Notice the different scale bars of the s-SNOM images. To quantify the field confinements, the FWHM of the line profiles are determined. In the cavity with the largest diameter of $D = 3\,\mu m$, the confinement is $\lambda_0/5.5$. The confinement can be improved by a factor of 7 by reducing the diameter of the cavity to $D = 1\,\mu m$, resulting in a field confinement of $\lambda_0/35$. Hence, reconfiguring the size of the cavities themselves allows for tuning the field confinement far below the diffraction limit of light.

Conventional fabrication techniques of SPhP resonators are associated with tremendous effort and precision. Accordingly, we highlight the advantage of direct laser writing targeted geometries by investigating more complex cavity geometries which can be directly written by applying laser pulses. Figure 5A displays the light microscope image of a fabricated square resonator with 3 μm edge length. The corresponding measured near-field images at varied frequencies are shown in Fig. 5B. The near-field amplitude obtained inside the resonator strongly depends on the chosen excitation frequency: Alternating patterns of enhanced and attenuated amplitude fields are shown for increasing excitation frequency.

Numerical field simulations of the out-of-plane component of the electric field are displayed in Fig. 5C. Remarkably, the experimentally obtained patterns are well reproduced, although the simulations exhibit more details because small imperfections of the laser-written

resonators disturb the optical near-field image. Highlighting the similarities, four dark regions at 900 cm$^{-1}$ are visible in the corners of the resonator for both, the simulated and experimentally measured data. In addition, at 910 cm$^{-1}$ the fields within the four corners of the resonator are strongly enhanced with a darker region in the center. Finally, a bright region of enhanced fields is shown in the center at an excitation frequency of 920 cm$^{-1}$.

Subsequently, we have investigated an exceptional resonator shape of a triangle with edge lengths of 2.7 μm (see light microscope image in Fig. 5D). The corresponding measured near-field amplitude images are shown in Fig. 5E for the same excitation frequencies investigated previously. Here, the lateral field extension inside the triangular resonator becomes more confined for increased frequency because the pattern transitions from a higher to a lower mode. The comparison with numerical field simulations (c.f. Figure 5F) reveals that the experimentally obtained data only match qualitatively with the simulations: both, the simulated and the measured field images, display a minimum in the center at 900 cm$^{-1}$. This pattern changes for increased excitation frequencies to a pronounced maximum within the center of the triangular resonator. Less pronounced details for the experimental data might be due to fabrication imperfections and roughness as an additional loss channel attenuating the observable pattern. A more detailed analysis of this deviation is beyond the scope of this paper and is a topic for future work.

## Discussion

In summary, we have demonstrated the potential of a promising platform for reconfigurable polariton photonics on the polar crystal SiC by employing the plasmonic phase-change material IST. Confined SPhP resonators are directly written and reconfigured inside IST to tune the mode confinement. The vast flexibility of our concept is shown by programming square and triangular resonators without cumbersome fabrication techniques such as electron beam lithography or focused ion beam milling.

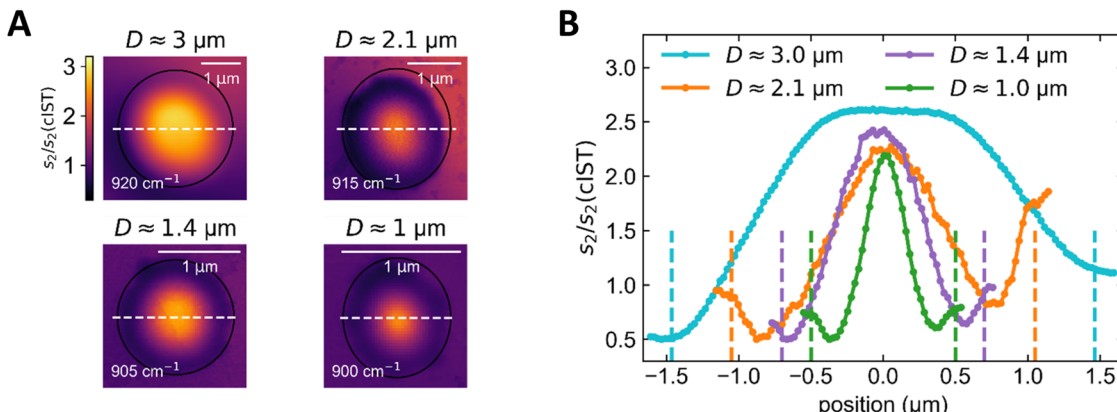

**Fig. 4 | Mode confinement of circular resonators with varied diameter. A** Optical near-field amplitude images of the (0,1) mode for different cavity diameters at the corresponding excitation frequencies. **B** Cross-section profile of the different cavities from A. Reducing the cavity diameter results in more confined fields. For the smallest cavity of 1.0 μm, a field confinement with a diameter of 320 nm is achieved.

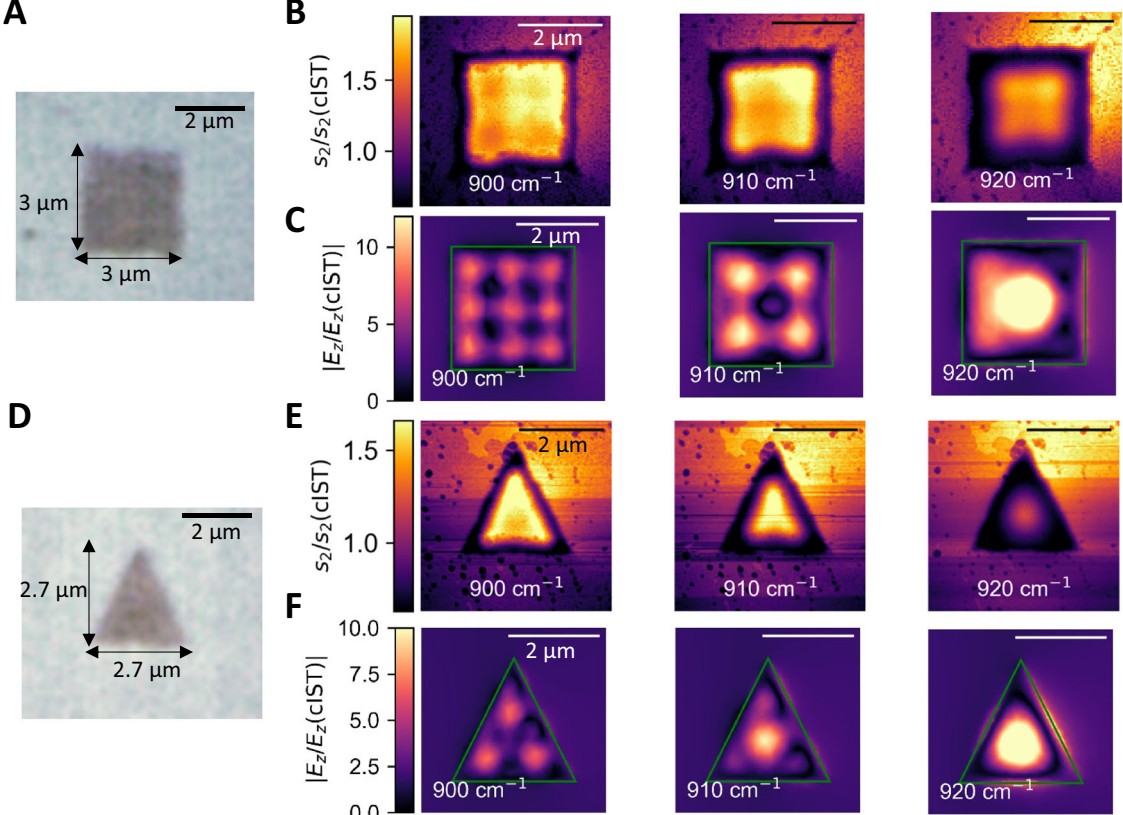

**Fig. 5 | Investigation of arbitrary resonator shapes. A** Light microscope image of an amorphous square resonator with a size of 3 × 3 μm². **B** Measured near-field amplitude images and numerical field simulations (**C**) of the out-of-plane component of the electric field at different excitation frequencies of 900 cm⁻¹, 910 cm⁻¹, and 920 cm⁻¹. Depending on the excitation frequency, the observed near-field pattern changes drastically. **D** Light microscope image of a triangle with 2.7 μm edge lengths. **E** Measured near-field amplitude images and numerical simulations (**F**) at the same excitation frequencies of (**B**). The scale bars equal 2 μm and the green shapes outline the simulated geometry.

Compared to previous similar studies[33,34] (see Supplementary Note 10 for a detailed comparative study), our work reveals similar performances with the unprecedented advantage of employing sophisticated spatially overlapping laser spots to define more complex structures such as squares and triangles. The polariton confinement could be further improved by reducing the film thickness of IST leading to even more confined resonance modes[34]. Moreover, the size of the resonators is independent of the size of the employed laser spot and can be tailored at will by combining spatially shifted crystallization and reamorphization pulses. Currently, reamorphization lead to some surface deformations of the capping layer as shown in Supplementary Note 11 and can be circumvented by a layerstack without capping[44,45] or optimizing the laser-induced phase change with the help of multi-physics simulations[46].

Since the plasmonic PCM IST is metallic in the entire infrared spectral range, the presented concept can be easily transferred to other polariton hosting materials. For example, surface plasmon polaritons could be investigated in highly doped semiconductors such

as CdO[47] with new widely-tunable laser sources[48] or hyperbolic phonon polaritons of 2d materials such as hexagonal boron nitride (hBN)[49] can be exploited with laser written IST structures. Here, a larger propagation length and therefore the observation of higher order modes could be achieved by combining the plasmonic PCM IST with isotopically pure hBN facilitating less losses[50–52]. We performed numerical simulations of amorphous IST cavities below 100 nm of hBN (see Supplementary Note 12) featuring enhanced field patterns inside the resonator and hyperbolic phonon polaritons launched by the crystalline edges outside of the resonators. Consequently, prestructuring the hBN into resonators is not required anymore, facilitating a convenient way for programmable polariton nanophotonics.

Even smaller cavities can be achieved by applying shorter wavelengths for switching the PCM (UV-lasers[53]), by increasing the numerical aperture or by exploiting more sophisticated spatially overlapping pulses, resulting in deep-subwavelength cavities with even more confined fields. Recently, Sheinfux et al. utilized strongly confined polariton resonators in hBN on nanopatterned holes inside a gold film in combination with bound state in continuum (BIC) interference to achieve tremendous quality factors up to 400[54]. This approach can be directly transferred to our demonstrated method, even allowing for reprogramming those cavities and fine-tuning of the investigated field confinements.

Another promising approach might be the programming of anisotropic SPhP resonators and launching structures by exploiting intrinsically anisotropic materials such as $\alpha$-MoO$_3$[55,56], V$_2$O$_5$[21], or $\beta$-Ga$_2$O$_3$[57] featuring a hyperbolic dispersion. These resonators would allow programming of complex non-symmetric resonance modes. We note that propagating polaritons on amorphous and crystalline IST below hBN and $\alpha$-MoO$_3$ have been independently shown in a recent publication without microstructuring the IST layer into resonators[58]. Hence, the ability of programming confined resonators within the plasmonic PCM IST boosts the research area of reconfigurable polariton optics and the performance is only limited by the applied polariton-hosting material.

Our work introduces a powerful platform for mid-IR reconfigurable polaritonic nanophotonics including metasurfaces[59,60], surface-enhanced infrared absorption (SEIRA)[12,61,62] and new sensor technologies with improved sensitivity.

## Methods
### Sample fabrication
A layer of amorphous In$_3$SbTe$_2$ and afterwards a capping layer of ZnS:SiO$_2$ were sputtered on a polished 6H-SiC substrate with direct current and radio frequency magnetron sputtering. The thicknesses of the IST layer and the ZnS:SiO$_2$ capping layer are determined to be 35 nm and 15 nm, respectively. The applied capping layer prevents the sample from oxidization and facilitates the crystallization of the IST layer because it works as an antireflection coating for the operating switching laser. Thus, more energy is coupled into the PCM layer[44].

### Optical switching
For the laser-induced phase transition, a home-built laser switching setup is used. The light of a laser diode with a wavelength of 660 nm was focused through a 10-fold objective with NA = 0.25 on the sample, which was placed on a Thorlabs piezo stage. For crystallization, 100 single pulses with a power of 80 mW and a pulse length of 3 μs were used to create a crystallized spot. Several spots were placed next to each other to obtain crystallized areas with an amorphous as-deposited region left out in the center. Reamorphization was achieved with a single pulse with a power of 300 mW and a pulse length of 20 ns.

### SNOM
Optical characterization was performed with a commercial scattering-type scanning near-field optical microscope by Neaspec GmbH

operated in pseudo-heterodyne detection mode[63]. To measure the local electric field, second demodulation order optical near-field amplitude $s_2$ and phase $\varphi_2$ were extracted. Topography was recorded with tapping-mode atomic force microscopy with a PtIr5-coated silicon tip by NanoWorld AG with a curvature radius ≈ 20 nm and resonance frequency ≈ 260 kHz. A tunable quantum cascade laser MIRcat-QT by Daylight Solutions Inc. in continuous wave mode was employed as a light source. The scan included an incident power of 3 mW to 7 mW, a tapping amplitude of about 120 nm (except for about 300 nm for the 3 μm cavity) and a scan speed of about 1.9 μm/s with a scan resolution of 25 nm/pixel.

### Simulations
Numerical simulations were performed with CST Studio Suite from Dassault Systèmes. Floquet port excitation with an incident angle of the p-polarized light of 60° normal to the surface and unit cell boundary conditions in lateral dimensions and open boundaries in vertical dimensions were assumed. The dielectric function of SiC was taken from Spitzer et al.[64]. For amorphous and crystalline IST, we applied the dielectric function shown in Supplementary Note 1, while for the capping layer ZnS:SiO$_2$ a constant refractive index of 2.1 is chosen. The electric field calculated with the frequency domain solver was extracted 1 nm above the surface and is normalized with respect to the electric field without amorphous cavities. The s-SNOM tip is not considered in the simulations because here the dominating launching mechanism relies on edge-launching. The accessible polariton wavevectors induced by the tip are significantly larger than the measured polariton wavevectors on SiC[65]. In contrast, the edges yield a broad spectrum of wavevectors with much lower accessible values. This has also been demonstrated in literature where observed polaritons can be perfectly reproduced by only assuming an edge-launched component[66,67].

## Data availability
The data that are necessary to interpret, verify and extend the research in the article are provided in the main text and/or SI. All the raw data files are available from the corresponding authors upon request.

## Code availability
The computer codes developed for this study are available from the corresponding author upon request.

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

## Acknowledgements

The authors thank Maike Kreutz for the sputter deposition of the thin film layer stack. The authors acknowledge support by the Deutsche Forschungsgemeinschaft (DFG No. 518913417 (T.T.) & SFB 917 "Nanoswitches" (T.T. and M.W.).

## Author contributions

L.C., K.W. and T.T. conceived the research idea; L.C., L.S. and K.W. designed the research; L.S. carried out the optical switching and the s-SNOM measurements; L.C. and L.S. analyzed the data and carried out the numerical simulations. M.W. provided the sputtering equipment and phase-change material expertise; all authors contributed to writing the manuscript.

## Funding

## Competing interests

The authors declare no competing interests.
