## [Peer Review File · Nature Communications]

Direct programming of confined Surface Phonon Polariton Resonators using the plasmonic Phase-Change Material In₃SbTe₂Reviewer #1 (Remarks to the Author):

Conrads et al. present a study on the reconfiguration of phonon polaritonic resonators achieved through laser-writing of the phase-change material In₃SbTe₂ (IST). The reversibility of the phase change, including designs with various shapes and sizes, is appealing and promising. The paper incorporates microfabrication, near-field imaging, far-field spectroscopy, numerical simulations, and analytical theory, communicated in a clear and articulate manner. However, despite the potential of this reconfiguration mechanism, a few concerns impede my acceptance of the paper in its current state for Nature Communications:

1. The microresonators' performance metrics, namely confinement of up to $\lambda/35$ and propagation length of around 1 micrometer, are not remarkable when compared to values previously documented in the literature. Other PhP resonators, as shown for instance in a publication by some of the present authors, exhibit superior features, including a confinement of up to $\lambda/70$ [Nat. Mater. 15, 870–875 (2016)]. Findings from other authors also demonstrate confinements exceeding $\lambda/35$ and propagation lengths surpassing 1 micrometer [arXiv:1905.02177v2 (2020); Adv. Mater. 34, 2104954 (2022)]. Could the authors conduct a comparative study to assess how their resonators perform relative to values reported in existing literature? Is there potential for improving already reported metrics through the fabrication technique detailed in this work? Incorporating the quality factor into the discussion could maybe provide valuable insights. [As a side note here, I'd also suggest the authors clearly explaining the methodology used to calculate the measured propagation length (1 μm).]
2. I am missing a more detailed discussion of the advantages of this technique for resonator fabrication, along with a direct comparison to analogous mechanisms reported previously, for example in [Nat. Mater. 15, 870–875 (2016)]. A description of the limitations, particularly in terms of resolution, potential sample degradation, sharpness of edges, constraints on resonator size, and the number of successful reconfigurations achievable, would be helpful. These factors directly impact the quality of the resonators, and I presume that some may be constrained by the size of the laser beam spot.
3. Moreover, the quality of fittings in Figure 2c could be improved. Currently, only a short segment of the profile is fitted, potentially leading to a substantial error. A more rigorous approach would involve a longer fit that encompasses multiple SPhP oscillations. Additionally, I suggest including corresponding error bars in Figure 2d.
4. The literature review concerning the tuning of PhPs appears to omit some pertinent works. I would recommend that the authors conduct a more exhaustive analysis of the existing literature.

Minor remarks:

Is the influence of the capping layer negligible on the calculations and simulations? This should be somehow justified.

The discrepancies between simulations and experimental results in Figure S7 are substantial. Could the authors improve this? For instance, adding a correcting factor that accounts for the broad angular distribution of the objective.

Reviewer #2 (Remarks to the Author):

In this manuscript, the authors demonstrate the non-volatile control of surface phonon polariton resonators using phase change materials (PCMs), In₃SbTe₂ (IST), on the silicon carbide crystal. Surface phonon polariton resonators are programmed by switching the state of IST from dielectric to

metal through laser irradiation. Confined resonant modes are investigated by scattering-type scanning near-field optical microscopy. The wavenumber of modes is shifted as the dimension of resonators changes. The concept of reconfigurable surface phonon polariton resonators is intriguing, and the programming approach employing plasmonic PCMs seems to be promising for manipulating surface phonon polaritons in diverse materials. The manuscript is clear and well-organized. However, it has several minor concerns that should be addressed before publication.

1. The authors investigate the mode confinement of resonators with varying diameter. I recommend that the authors provide the simulated out-of-plane component of the electric field for each differently sized resonator. It can be helpful for readers to understand the relation between the size of resonators and frequencies.
2. Topography of resonators is measured by atomic force microscopy. I wonder why the visibility of the resonators is too low to recognize the boundary between amorphous and crystalline states. Is there any method to distinguish the state of plasmonic PCMs directly, as shown in Figure 5A and 5D?
3. I suggest that the authors add an explanation for the role of the capping layer of ZnS:SiO₂.
4. Experimental methods lack sufficient detail. The authors should provide a more comprehensive description of the methods, especially numerical simulations.
5. In Figure S7, experimental reflectance (B) and simulated reflectance (C) appear different. Reasons for the discrepancy should be described in the manuscript.

Overall, I would recommend the publication of this work in Nature Communications after these minor concerns have been addressed.

Reviewer #3 (Remarks to the Author):

This is an interesting paper presenting new results on experimentally reconfiguring surface phonon polaritons (SPhPs) resonators using newly introduced plasmonic phase change material IST. The authors have already documented their ability to reconfigurable control of IST as an ideal programmable nanophotonics platform in several publications. Here they focus on SPhP resonators within an IST thin film and attempt to investigate how resonance modes were tuned by multiple cavity geometries directly written by laser pulses. Overall, it is an interesting idea and the manuscripts contains well-analyzed results, but I think the novelty shown in the current manuscript is compromised by several aspects, as detailed below. I find a revised version might be appropriate for Nature Communications.

Issues:

1. The authors choose the optical writing of confined SPhP resonators inside an amorphous IST layer, where multiple crystallizing laser pulses were applied spatially next to each other and an amorphous as-deposited region is left out in the center. This way of fabrication sounds cumbersome. It would be more intuitive to start from a thermally crystallized IST layer, then applying amorphizing laser pulses to create SPhP resonators. Would this approach create similar deformation (cracking the capping) as shown in Supplementary Note 8? If it is out of technical reason to carry out experiments not in this way, the authors should elaborate more on this in the main text, because this hampers the degree of flexibility of using IST films.
2. The concept of using a thin film of PCM on a polar crystal to tune SPhPs is similar to what have been demonstrated in authors' previous works, e.g., Li et al. 15, 870 (2016) and Sumikura et al. 19, 2549 (2019). It would be much more convincing if the authors first put forward comparisons of the

new idea in the present work with these.

3. In Fig. 5, the agreement between measured near-field amplitude images and numerical field simulations is not satisfactory. The authors ascribe it to "fabrication imperfections and roughness". Did the authors analyze the detailed crystal morphology around the cIST/aIST boundary with other technique such as SEM? This may help to clarify the underlying mechanism.

Direct programming of confined Surface Phonon Polariton Resonators with the plasmonic Phase-Change Material In_3SbTe_2

Responses to the reviewers' comments and a summary of the changes made to the revised manuscript:

We would like to thank the editor and the reviewers for their review of our work. In this letter, we provide point-by-point responses to each reviewer's comments. The implemented changes are highlighted in the revised marked copy of the manuscript.

Reviewer #1

Reviewer's general statement:

Conrads et al. present a study on the reconfiguration of phonon polaritonic resonators achieved through laser-writing of the phase-change material In_3SbTe_2 (IST). The reversibility of the phase change, including designs with various shapes and sizes, is appealing and promising. The paper incorporates microfabrication, near-field imaging, far-field spectroscopy, numerical simulations, and analytical theory, communicated in a clear and articulate manner. However, despite the potential of this reconfiguration mechanism, a few concerns impede my acceptance of the paper in its current state for Nature Communications:

Our response:

We thank the reviewer for evaluating our concept as appealing and promising and highlighting the clear structure of the manuscript. We addressed the raised concerns in order to improve the quality of our work.

Reviewer's comment 1:

The microresonators' performance metrics, namely confinement of up to $\lambda/35$ and propagation length of around 1 micrometer, are not remarkable when compared to values previously documented in the literature. Other PhP resonators, as shown for instance in a publication by some of the present authors, exhibit superior features, including a confinement of up to $\lambda/70$ [Nat. Mater. 15, 870–875 (2016)]. Findings from other authors also demonstrate confinements exceeding $\lambda/35$ and propagation lengths surpassing 1 micrometer [arXiv:1905.02177v2 (2020); Adv. Mater. 34, 2104954 (2022)]. Could the authors conduct a comparative study to assess how their resonators perform relative to values reported in existing literature? Is there potential for improving already reported metrics through the fabrication technique detailed in this work? Incorporating the quality factor into the discussion could maybe provide valuable insights. [As a side note here, I'd also suggest the authors clearly explaining the methodology used to calculate the measured propagation length (1 μm).]

Our response:

The reviewer is correct that the field confinement reported in this work is not exceeding those of compared to similar work. However, the unprecedented advantage of our presented technique relies in reconfiguring already written resonators to modify the field confinement afterwards. To our best knowledge, this has not been demonstrated yet. In addition, we anticipate that an IST layer with a reduced thickness will lead to even more confined resonance modes as already demonstrated by Li et al. (Nat. Mater. 15, 870–875 (2016)) for 7 nm thin films. Actually, their reported polariton confinement for thicker layers was in the same range as our values. To support this claim, we calculated the SPhP dispersion for a thinner layer of IST which clearly displays more confined SPhPs upon decreasing the film thickness (see Figure S3.3 in Supplementary Note 3).

We appreciate the suggestion of taking the quality factor into consideration and determined for the (0,1) modes of the circular cavities the quality factors by fitting a Lorentzian function to the measured s-SNOM amplitude signals for varied frequencies.

The propagation length of the propagating polaritons is in the range of several micrometers and is only an approximation by how many SPhP fringes are visible.

Actions taken:

- We added a comparative study in the Supporting Information:

Supplementary Note 10: Comparative Study with previous literature

Finally, we performed a comparative study with previous related work in literature to evaluate the different strengths of our method compared to other work. We choose several characteristics such as reconfigurability, field confinement and quality factor. The results are displayed in Table S2.

Table S2. Comparative study of polariton resonators

Publication	material	Near-/Farfield	Prop. length	Field confinement Quality factor	Mode analysis	Reconfigurability
This work	SiC	Both	~ 1 μm	$\lambda/35$ Q ~ 50	yes	Modify shapes
Li et al. ^[10]	SiO ₂	Both	~ 1 μm	$\lambda/30$ for d=30 nm $\lambda/80$ for d=7 nm Q ~ 95	yes	Erase and write
Sumikura et al. ^[2]	SiC	Nearfield	~ 1 μm	$\lambda/50$	yes	Turn on and off
Wang et al. ^[15]	SiC	Both	-	Q ~ 50-70	yes	No
Tamagnone et al. ^[16]	hBN	Nearfield	-	Q ~ 300	yes	No
	α -MoO ₃	Both	-	Q ~ 250	no	No
Duan et al. ^[17]	hBN	Nearfield	-	$\lambda/42$ Q ~ 165	yes	graphene doping
Folland et al. ^[18]	hBN	Nearfield	~ 1-3 μm	-	-	Heating VO ₂
Sheinfux et al. ^[19]	hBN	Nearfield	-	Q ~ 50-480	yes	No

Overall, our optically written resonators show similar performance compared to previous work in literature based on bulk polar crystals like SiC with the unprecedented advantage of easy reconfigurability to modify the resonator shapes to tune the field confinement. We anticipate even better performance by combining our concept with thinner IST films or low-loss 2d materials such as hBN or α -MoO₃.

- More details to improve those metrics is added in the discussion in the main text:

Compared to previous similar studies^[30,31] (see **Supplementary Note 10** for a detailed comparative study), our work reveals similar performances with the unprecedented advantage of employing sophisticated spatially overlapping laser spots to define more complex structures such as squares and triangles. The polariton confinement could be improved by reducing the film thickness of IST leading to even more confined resonance modes^[31].

- We included the quality factor within our analysis:

The quality factor Q is determined by fitting a Lorentzian function and reveals values up to 50 (see Supplementary Note 7) in accordance with literature values of similar systems^[6,24] and can be further improved by reducing the IST film thickness.

Supplementary Note 7: Quality factor of the resonators

The quality factor of the resonators is determined by fitting a Lorentzian function to the optical near-field amplitude contrasts at varied excitation frequencies for the different cavity diameters (see solid curves in **Figure S7**). Due to the strong asymmetry of the measured data points, only two points left to the maximum are taken into account. The determined quality factors are in the range of 50, except for the orange curve which shows a quality factor of 31.

Figure S7: Fitted Lorentzian functions (solid curves) to the experimentally obtained near-field amplitude contrasts. The obtained quality factors for $D = 1.0 \mu\text{m}$, $1.4 \mu\text{m}$, $2.1 \mu\text{m}$, and $3.0 \mu\text{m}$ are 50, 46, 31, and 42, respectively.

- We added the calculated polariton confinement for different IST layer thicknesses in Supplementary Note 3:

Figure S3.3: Calculated polariton confinement for different IST layer thicknesses.

Another possibility to increase the polariton confinement is reducing the IST layer thickness (see **Figure S3.3**), revealing values up to 125 for a 7 nm thin IST layer.

Reviewer's comment 2:

I am missing a more detailed discussion of the advantages of this technique for resonator fabrication, along with a direct comparison to analogous mechanisms reported previously, for example in [Nat. Mater. 15, 870–875 (2016)]. A description of the limitations, particularly in terms of resolution, potential sample degradation, sharpness of edges, constraints on resonator size, and the number of successful reconfigurations achievable, would be helpful. These factors directly impact the quality of the resonators, and I presume that some may be constrained by the size of the laser beam spot.

Our response:

The reviewer is correct that a more detailed description of the applied mechanism is missing so far. In fact, mechanism of optically writing resonator structures within a PCM is not new and has been reported previously for the commonly used dielectric PCM GST (Li et al. (Nat. Mater. 15, 870–875 (2016)); Sumikura et al. (Nano Letters 19, 4 (2019))). However, defining more complex resonators structures such as square and triangular shapes via laser irradiation has not been reported yet. We take advantage of sophisticated spatially overlapping elliptically shaped laser spots to obtain these other shapes. Another distinct difference is the used novel plasmonic PCM IST compared to the dielectric PCM GST. Previous work demonstrated that the reflection phase varies dependent on thickness of the metal on top of the PCM (Sumikura et al. (Nano Letters 19, 4 (2019))). This behavior drastically changes in our case where the metallic behavior of crystalline IST leads to a well-defined reflection phase of $-\pi$ in accordance with the expectations.

In the following, we address the various limitations mentioned by the reviewer:

- a) **Resolution:** The resolution of optically writing resonators can be precisely adjusted with piezoelectric actuators to control the position of the laser spot in the nanometer range.
- b) **Sample degradation:** Currently, reamorphization lead to some surface degradation of the capping layer as visible in Figure S11 (see Supplementary Note 11), caused by the strong heat induced in the layerstack. Accordingly, the capping layer above the PCM displays some cracks (PhD Thesis Andreas Heßler RWTH Aachen University 2022). It was found that this can be circumvented by applying a more sophisticated layerstack with a thicker capping layer or by removing the capping layer entirely (Barnett et al., Nano Letters 21, 9012-9020, 2021) with the disadvantage of possible oxidization. However, finding the best suited layerstack with tailored thicknesses of the applied materials is a current research topic and must be optimized for the specific case. Here, multiphysics simulations provide a powerful tool for exploring the optimal layerstack with tailored heat distribution and crystallization kinetics (Meyer et al., Nanophotonics 9, 3, 2020).
- c) **Sharpness of edges:** The sharpness of the crystallized resonator edges is a topic of future work, including how this influences the polariton launching efficiency. In our case, we assume relatively sharp edges since the derived reflection phase shows a constant behavior. This is in good agreement with previous work where much smaller laser powers for crystallization already resulted in fully crystallized PCM layers (Conrads et al. ACS Nano 2023; Michel et al. Adv. Mater. 2019; Conrads et al. Adv. Opt. Mat. 2023).
- d) **Constraints of resonator size:** The resolution of optically writing resonators is not limited by the size of the switching laser. Combining sophisticated spatially overlapping reamorphization and crystallization pulses allows for programming structures well below the size of the applied laser spot down to a few hundred nanometers and can be even more reduced (Heßler et al. ACS Photonics 9,5 (2022)).
- e) **Switching cycles:** Several successful switching cycles (up to 20) via laser irradiation has been demonstrated by Heßler et al. (Nature Communications 12, 1, 2021).

Actions taken:

- We extended the description of our employed method in the discussion section:

Compared to previous similar studies^[30,31] (see **Supplementary Note 10** for a detailed comparative study), our work reveals similar performances with the unprecedented advantage of employing sophisticated spatially overlapping laser spots to define more complex structures such as squares and triangles. The polariton confinement could be further improved by reducing the film thickness of IST leading to even more confined resonance modes^[31]. Moreover, the size of the resonators is independent from the size of the employed laser spot and can be tailored at will by combining spatially shifted crystallization and reamorphization pulses. Currently, reamorphization lead to some surface deformations of the capping layer as shown in **Supplementary Note 11** and can be circumvented by a layerstack without capping^[40,41] or optimizing the laser induced phase-change with the help of multiphysics simulations.^[42]

- We added a description about resolution and resonator size in Supplementary Note 4:

For programming these resonators, the size is not limited by the spot size, but rather by the step size (accuracy/resolution) of the applied piezo actuators used for positioning the sample and consequently in the nanometer range. It is even possible by combining sophisticated spatially overlapping crystallization and reamorphization pulses to tailor the size of the resonators at will down to a few hundred nanometers.^[11]

Reviewer's comment 3:

Moreover, the quality of fittings in Figure 2c could be improved. Currently, only a short segment of the profile is fitted, potentially leading to a substantial error. A more rigorous approach would involve a longer fit that encompasses multiple SPhP oscillations. Additionally, I suggest including corresponding error bars in Figure 2d.

Our response:

One might think that a longer fit segment would improve the results about the polariton wavevector. Unfortunately, at small excitation frequencies topographical artifacts such as scratches on the sample lead to the observed features in the near-field amplitude. At the trench with a lower topography, the SNOM tip is closer to the IST layer leading to more pronounced near-fields detected by the SNOM and therefore causing peaks in the observed near-field amplitude. The propagating SPhPs are not the reason for these peaks and therefore the segment for fitting is reduced. To further prove that the observed peaks in the near-field amplitude are caused by topographical artifacts, we added the line profile of the topography in the Supplementary Note 2. These peaks perfectly coincide the topographical artifacts. Moreover, we recorded a second data set of propagating SPhPs which are now also included in Supplementary Note 2.

Also, we followed the suggestion of the reviewer and included error bars within the dispersion plot. The error bars are taken from the fitting, which are very small and therefore nearly not visible in the dispersion plot.

Actions taken:

- We slightly increased the range of the fitting for the polariton line profiles, but omitted topographical artifacts for small excitation frequencies. Moreover, we included error bars within the polariton dispersion diagram:

Figure 2: Propagating SPhPs launched by a cIST edge. [...]

- We added topographic line profiles and an additional data set in Supplementary Note 2:

In the zoom-in to the fits from Figure S2.2 more peaks appear especially for small frequencies at 880 cm^{-1} and 890 cm^{-1} . These peaks are omitted from the polariton fit since they are only topographical artifacts. The near-field amplitude line profiles and the corresponding topography line profile averaged over 10 lines are displayed together with the topography image in Figure S2.3. Strong minima in the topography caused by scratches at 1.8 μm , 2.2 μm and 2.6 μm exactly coincide with narrow peaks in the near-field amplitude. At the trenches with a lower topography, the SNOM tip is closer to the IST layer leading to more pronounced near-fields detected by the SNOM and therefore causing peaks in the observed near-field amplitude. Consequently, we assign those features to measurement artifacts which are not caused by polaritons.

Figure S2.3: Topography line profile (black line) in combination with near-field amplitude line profiles (colored lines). Strong minima in the topography exactly coincide with narrow peaks in the near-field amplitude (highlighted with red dashed lines).

Figure S2.4: Additional data set of propagating SPhPs. **A)** Topography image of another crystalline IST edge. Scratches and other particles disturb the obtained image. **B)** Measured near-field amplitude images at varied excitation frequencies. **C)** Extracted line profiles and the fit of the damped harmonic oscillation. **D)** Calculated polariton dispersion with the experimentally obtained polariton wavevectors.

Reviewer's comment 4:

The literature review concerning the tuning of PhPs appears to omit some pertinent works. I would recommend that the authors conduct a more exhaustive analysis of the existing literature.

Our response:

We thank the reviewer for highlighting missing pertinent work. We reworked our literature basis accordingly.

Actions taken:

- The following references are now included:

[15] Y. Wu, J. Duan, W. Ma, Q. Ou, P. Li, P. Alonso-González, J. D. Caldwell, Q. Bao, *Nature Reviews Physics* **2022**, *4*, 578.

[16] A. D. Dunkelberger, C. T. Ellis, D. C. Ratchford, A. J. Giles, M. Kim, C. S. Kim, B. T. Spann, I. Vurgaftman, J. G. Tischler, J. P. Long, O. J. Glembocki, J. C. Owrutsky, J. D. Caldwell, *Nature Photonics* **2018**, *12*, 50.

[17] S. Dai, Q. Ma, M. K. Liu, T. Andersen, Z. Fei, M. D. Goldflam, M. Wagner, K. Watanabe, T. Taniguchi, M. Thiemens, F. Keilmann, Janssen, G. C. A. M., S.-E. Zhu, P. Jarillo-Herrero, M. M. Fogler, D. N. Basov, *Nature Nanotechnology* **2015**, *10*, 682.

[18] J. Duan, F. J. Alfaro-Mozaz, J. Taboada-Gutiérrez, I. Dolado, G. Álvarez-Pérez, E. Titova, A. Bylinkin, A. I. F. Tresguerres-Mata, J. Martín-Sánchez, S. Liu, J. H. Edgar, D. A. Bandurin, P. Jarillo-Herrero, R. Hillenbrand, A. Y. Nikitin, P. Alonso-González, *Adv. Mater.* 2022, 34, 2104954.

[19] X. Yang, F. Zhai, H. Hu, D. Hu, R. Liu, S. Zhang, M. Sun, Z. Sun, J. Chen, Q. Dai, *Adv. Mater.* 2016, 28, 2931.

[20] N. A. Aghamiri, G. Hu, A. Fali, Z. Zhang, J. Li, S. Balendhran, S. Walia, S. Sriram, J. H. Edgar, S. Ramanathan, A. Alù, Y. Abate, *Nature Communications* 2022, 13, 4511.

[21] J. Taboada-Gutiérrez, G. Álvarez-Pérez, J. Duan, W. Ma, K. Crowley, I. Prieto, A. Bylinkin, M. Autore, H. Volkova, K. Kimura, T. Kimura, M.-H. Berger, S. Li, Q. Bao, X. P. A. Gao, I. Errea, A. Y. Nikitin, R. Hillenbrand, J. Martín-Sánchez, P. Alonso-González, *Nature Materials* 2020, 19, 964.

[22] M. Chen, X. Lin, T. H. Dinh, Z. Zheng, J. Shen, Q. Ma, H. Chen, P. Jarillo-Herrero, S. Dai, *Nature Materials* 2020, 19, 1307.

Reviewer's comment 5: Minor remarks from Reviewer:

Is the influence of the capping layer negligible on the calculations and simulations? This should be somehow justified.

The discrepancies between simulations and experimental results in Figure S7 are substantial. Could the authors improve this? For instance, adding a correcting factor that accounts for the broad angular distribution of the objective.

Our response:

We thank the reviewer for raising the point about the capping layer. We included the influence of the capping layer in all calculations and simulations, since the influence is not negligible, leading to a modified polariton dispersion relation.

In addition, we reworked the numerical simulations for the measured FTIR spectra to achieve a better agreement between simulations and experimental data. This was done by modifying the simulated unit cell and considering multiple resonators with slightly different diameters to take small fabrication deviations into account.

Actions taken:

- We included calculated dispersion diagrams for different capping layer thicknesses within Supplementary Note 3:

The higher effective permittivity of the capping layer compared to air shifts the dispersion to lower frequencies. This effect becomes even more pronounced if the thickness of the capping layer is increased (see the calculated dispersion curves for varied capping layer thickness in **Figure S3.2**).

Figure S3.2: Calculated SPhP dispersion for varied capping thicknesses of 0 (A) and 50 nm (B)

- We reworked the numerical simulations of the farfield spectra:

To improve the agreement between experiment and simulations, we modified the simulated unit cell of the SPhP resonators. In particular, we simulated three resonators within the same unit cell with slightly different diameters to take fabrication imperfections into account which led to broader resonances as visible in the measured reflectance spectra.

Figure S8: Far-field measurements and simulations of cavity arrays. [...]

Reviewer #2

Reviewer's general statement:

In this manuscript, the authors demonstrate the non-volatile control of surface phonon polariton resonators using phase change materials (PCMs), In₃SbTe₂ (IST), on the silicon carbide crystal. Surface phonon polariton resonators are programmed by switching the state of IST from dielectric to metal through laser irradiation. Confined resonant modes are investigated by scattering-type scanning near-field optical microscopy. The wavenumber of modes is shifted as the dimension of resonators changes. The concept of reconfigurable surface phonon polariton resonators is intriguing, and the programming approach employing plasmonic PCMs seems to be promising for manipulating surface phonon polaritons in diverse materials. The manuscript is clear and well-organized. However, it has several minor concerns that should be addressed before publication.

Our response:

We thank the reviewer for his minor concerns and will address them in the following.

Reviewer's comment 1:

The authors investigate the mode confinement of resonators with varying diameter. I recommend that the authors provide the simulated out-of-plane component of the electric field for each differently sized resonator. It can be helpful for readers to understand the relation between the size of resonators and frequencies.

Our response:

We agree with the reviewer that the simulated out-of-plane component of the electric field provides valuable insights in the mode behavior between the size of the resonator and the frequencies.

Actions taken:

- We included a short notice about the numerical simulations in the manuscript (page 6):

Moreover, we performed numerical field simulations of the out-of-plane electric field component resembling the experimentally observed field patterns very well (see **Supplementary Note 6**).

- Field simulations for all cavity diameters and frequencies are now added in the Supporting Information.

Supplementary Note 6: Field simulations of the cavities with varied diameters

Figure S6: Numerical field simulations of the out-of-plane electric field for varied cavity diameters and excitation frequencies.

Reviewer's comment 2:

Topography of resonators is measured by atomic force microscopy. I wonder why the visibility of the resonators is too low to recognize the boundary between amorphous and crystalline states. Is there any method to distinguish the state of plasmonic PCMs directly, as shown in Figure 5A and 5D?

Our response:

The reviewer is correct that the boundary between amorphous and crystalline IST is barely visible in the topography images. It has been previously shown that upon crystallization the density of the PCM increases by 8-10 % resulting in a total height difference of only a few nanometers. Since the PCM is also covered by the capping layer, the topography contrast is furthermore reduced. Without a capping layer, the topography contrast between the two phases would be clearly distinguishable. A well-suited method to determine the different states of the PCM is given by optical light microscope images, where crystalline IST appears very bright in contrast.

Actions taken:

- We added light microscope images of the circular resonators in the Supporting Information and included a sentence in the main manuscript about this issue.

The corresponding s-SNOM amplitude images for the different cavity diameters with additional light microscope images allowing for a better distinction between the crystalline and amorphous phase can be found in **Supplementary Note 5**.

Figure S5.2: Light microscope images of the different cavities. The crystallized IST appears bright while the amorphous IST is significantly darker. The scale bars equal 3 μm .

Reviewer's comment 3:

I suggest that the authors add an explanation for the role of the capping layer of ZnS:SiO₂.

Our response:

The capping layer on top of the PCM layer fulfills different tasks. First, it prevents the sample from oxidization. Second, the crystallization with a capping layer is easier since it works as an antireflection coating for the operating switching laser. Thus, more energy is coupled into the PCM layer upon laser irradiation (see PhD Thesis Andreas Heßler, RWTH Aachen University 2022).

Actions taken:

- An explanation about the role of the capping layer is added in the Methods section of the manuscript.

The applied capping layer prevents the sample from oxidization and facilitates the crystallization of the IST layer because it works as an antireflection coating for the operating switching laser. Thus, more energy is coupled into the PCM layer.^[40]

Reviewer's comment 4:

Experimental methods lack sufficient detail. The authors should provide a more comprehensive description of the methods, especially numerical simulations.

Our response:

We thank the reviewer for this point and reworked the experimental methods section.

Actions taken:

- We reworked the methods section and provided more comprehensive information:

Simulations:

Numerical simulations were performed with CST Studio Suite from Dassault Systèmes. Floquet port excitation with an incident angle of the p-polarized light of 60° normal to the surface and unit cell boundary conditions in lateral dimensions and open boundaries in vertical dimensions were assumed. The dielectric function of SiC was taken from Spitzer et al.^[57]. For amorphous and crystalline IST, we applied the dielectric function shown in Supplementary Note 1, while for the capping layer ZnS:SiO₂ a constant refractive index of 2.1 is chosen. The electric field calculated with the frequency domain solver was extracted 1 nm above the surface and is normalized with respect to the electric field without amorphous cavities. The s-SNOM tip is not considered in the simulations.

Reviewer's comment 5:

In Figure S7, experimental reflectance (B) and simulated reflectance (C) appear different. Reasons for the discrepancy should be described in the manuscript.

Our response:

We agree with the reviewer about the discrepancies in the experimental and simulated reflectance spectra. We attribute this to small fabrication imperfections across the laser-switched resonator arrays, leading to broader resonances in the experiment compared to the simulations. Consequently, we modified the simulated unit cell and included three resonators with slightly different diameters in the unit cell. Now, the simulated spectra match much better with the experimentally obtained ones.

Actions taken:

- We reworked the numerical simulations in Supplementary Note 8:

Figure S8: Far-field measurements and simulations of cavity arrays. [...]

Reviewer #3

Reviewer's general statement:

This is an interesting paper presenting new results on experimentally reconfiguring surface phonon polaritons (SPhPs) resonators using newly introduced plasmonic phase change material IST. The authors have already documented their ability to reconfigurable control of IST as an ideal programmable nanophotonics platform in several publications. Here they focus on SPhP resonators within an IST thin film and attempt to investigate how resonance modes were tuned by multiple cavity geometries directly written by laser pulses. Overall, it is an interesting idea and the manuscripts contains well-analyzed results, but I think the novelty shown in the current manuscript is compromised by several aspects, as detailed below. I find a revised version might be appropriate for Nature Communications.

Our response:

We appreciate reviewer for valuing our work as interesting and recommending a revised version for Nature Communications. We addressed the suggestions from the reviewer in the following.

Reviewer's comment 1:

The authors choose the optical writing of confined SPhP resonators inside an amorphous IST layer, where multiple crystallizing laser pulses were applied spatially next to each other and an amorphous as-deposited region is left out in the center. This way of fabrication sounds cumbersome. It would be more intuitive to start from a thermally crystallized IST layer, then applying amorphizing laser pulses to create SPhP resonators. Would this approach create similar deformation (cracking the capping) as shown in Supplementary Note 8? If it is out of technical reason to carry out experiments not in this way, the authors should elaborate more on this in the main text, because this hampers the degree of flexibility of using IST films.

Our response:

First of all, we would like to thank the reviewer for raising this point. We agree that our applied fabrication method seems cumbersome, and it would be more intuitive to start from a thermally crystallized IST layer. However, we faced some problems in crystallizing IST thermally on substrates different from conventional silicon. Moreover, we only had a limited amount of SiC substrates and therefore thermal crystallization was not suitable since we could not test different thermal crystallization parameters. Therefore, we applied laser crystallization since we are very experienced in spatially tailored crystallization with laser spots and we could test various laser parameters to achieve the most homogeneous areas (Conrads et al. ACS Nano 17, 24, 2023). Indeed, we expect that the observed cracks within the capping layer upon reamorphization would also appear in a thermally crystallized IST film due to the strong heat introduced in the material, leading to an increase in the volume and hence the formation of cracks within the capping layer. This can be circumvented by removing the capping layer (Barnett et al. Nano Letters 21, 9012-9020, 2021) or applying a thicker capping layer which would then attenuate the observed amplitude signals from the polaritons (see also Reviewer 1 comment 2). However, finding the best suited layerstack with tailored thicknesses of the applied materials is an open research topic and must be optimized for the specific case. Here, multiphysics simulations can provide a powerful tool for exploring the optimal layerstack with tailored heat distribution and crystallization kinetics (Meyer et al., Nanophotonics 9, 3, 2020).

Actions taken:

- We added an explanation for the applied method in Supplementary Note 4:

We chose our applied method to fabricate the resonator structures instead of starting from a thermally crystallized IST layer and applying reamorphization pulses because spatially tailoring crystallization with laser spots offers a larger degree of freedom to test various crystallization parameters. Moreover, the fixed amount of SiC substrates limited our possibilities to investigate a test series of different thermal crystallization parameters for homogeneous layers.

Reviewer's comment 2:

The concept of using a thin film of PCM on a polar crystal to tune SPhPs is similar to what has been demonstrated in authors' previous works, e.g., Li et al. 15, 870 (2016) and Sumikura et al. 19, 2549 (2019). It would be much more convincing if the authors first put forward comparisons of the new idea in the present work with these.

Our response:

The reviewer is correct that using a thin film of PCM in combination with a polar crystal to tune SPhPs is not new and has been demonstrated previously for the commonly used dielectric PCM GST. However, defining more complex resonator structures such as square and triangular shapes via laser irradiation has not been reported yet. We take advantage of sophisticated spatially overlapping elliptically shaped laser spots to obtain these other shapes. Another distinct difference is the investigated material. In contrast to conventional dielectric PCMs only featuring a contrast in the refractive index, IST switches to a metallic ($\epsilon < 0$) crystalline phase. While previous work (Li et al. (Nat. Mater. 15, 870–875 (2016)); Sumikura et al. (Nano Letters 19, 4 (2019))) demonstrated that the reflection phase varies dependent on thickness of the metal on top of the PCM. This behavior

drastically changes in our case where the metallic behavior of crystalline IST leads to a well-defined reflection phase of $-\pi$ in accordance with the expectations.

Actions taken:

- We included a comparative study in Supplementary Note 10 and highlight the advantage of directly reconfiguring the resonator shapes in the discussion section of the main manuscript.

Compared to previous similar studies^[30,31] (see **Supplementary Note 10** for a detailed comparative study), our work reveals similar performances with the unprecedented advantage of employing sophisticated spatially overlapping laser spots to define more complex structures such as squares and triangles.

Reviewer's comment 3:

In Fig. 5, the agreement between measured near-field amplitude images and numerical field simulations is not satisfactory. The authors ascribe it to "fabrication imperfections and roughness". Did the authors analyze the detailed crystal morphology around the cIST/aIST boundary with other technique such as SEM? This may help to clarify the underlying mechanism.

Our response:

For the reviewer the agreement between measured near-field amplitude images and numerical field simulations in Figure 5 seems not to be satisfactory. Therefore, we have to better highlight the similarities between both: For the square resonator at 900 cm^{-1} four dark regions in the corners are distinguishable. Moreover, at increased excitation frequency at 910 cm^{-1} the fields in the four corners are strongly enhanced with a darker region in the center of the resonator. This behavior changes at 920 cm^{-1} where the fields are mostly enhanced in the center. For the triangular resonator and at the smallest excitation frequency at 900 cm^{-1} , the center of the resonator exhibits a darker region of less pronounced fields, while at the corners the fields are enhanced. Increasing the excitation frequency results in enhanced fields at the center of the resonator. All in all, the simulations show more detailed features compared to the experiment which are attributed to the perfect resonator shape assumed in the simulations.

A detailed study about the crystal morphology and the influence of the boundary are not in the scope of this paper and a topic for future investigations. Our work clearly demonstrates programming of complex resonator shapes within a PCM and even allowing for post-fabrication adaptations to modify the mode structure and field confinement.

Actions taken:

- We added a more detailed description highlighting the similarities of the shown images:

Highlighting the similarities, four dark regions at 900 cm^{-1} are visible in the corners of the resonator for both, the simulated and experimentally measured data. In addition, at 910 cm^{-1} the fields within the four corners of the resonator are strongly enhanced with a darker region in the center. Finally, a bright region of enhanced fields is shown in the center at an excitation frequency of 920 cm^{-1} . [...]

Both [for the triangular resonator], the simulated and the measured field images, display a minimum in the center at 900 cm^{-1} . This pattern changes for increased excitation frequencies to a pronounced maximum within the center of the triangular resonator. Less pronounced details for the experimental data might be due to fabrication imperfections and roughness as an additional loss channel attenuating the observable pattern. A more detailed analysis about this deviation is beyond the scope of this paper and is a topic for future work.

Reviewer #1 (Remarks to the Author):

I would like to thank the authors for their detailed response.

On the technical side, I think the paper is instructive, rigorous and clear. I believe that just one concern has not been solved yet: the agreement between the simulations and the experiments in Fig. 5. I recommend improving this. I think that considering the tip in the simulations is important. There are well-known and standard methods to do this.

I have some concerns regarding the novelty of the work. First, there have been previous reports on reconfigurability using IST, many of them by some coauthors of the present work, so the technique/materials are not really novel. Second, as shown by the authors, the performance metrics of the resonators are comparable to those in found in the existing literature, and some of these prior works also offer tuning capabilities, at times even in operando. Third, it seems that further refinement of the technique is necessary: reamorphization, sharpness of the edges, optimizing the layerstack and the laser induced phase-change, etc. While I acknowledge that there is potential in the concept, such potential appears to be underexplored and underexploited in the current manuscript. My concern is that modifying the confinement, fabricating squares and triangles, and simulating an hBN layer on top of some circles drawn in IST may not demonstrate the full potential of the technique and therefore may be insufficient for publication in Nature Communications. However, I would be happy to review a revised version of the manuscript if these aspects are addressed.

Minor remark: the references to the papers to twisted polaritonic materials should be revised. Refs 20, 21 are not really about doping.

Reviewer #2 (Remarks to the Author):

In this work, the authors highlight controlling the surface phonon polariton resonator with phase change materials In_3SbTe_2 . Numerical simulations of the out-of-plane electric field are conducted and added in Supplementary Note 6. Optical microscopy images of both crystallized and amorphous IST are shown in Figure S5, which can be helpful for readers. The authors add a detailed explanation for the capping layer and the discrepancy between experimental data and simulation data in Figure S7. A more comprehensive description of the experimental methods is provided. The previous concerns have been well-revised and the manuscript has been significantly improved. Therefore, I recommend its publication in Nature Communications.

Reviewer #3 (Remarks to the Author):

The authors have addressed my comments. I recommend the manuscript for publication.

Response Letter

Reviewer #1:

I would like to thank the authors for their detailed response.

On the technical side, I think the paper is instructive, rigorous and clear. I believe that just one concern has not been solved yet: the agreement between the simulations and the experiments in Fig. 5. I recommend improving this. I think that considering the tip in the simulations is important. There are well-known and standard methods to do this.

Our response:

We thank the reviewer for evaluating our paper as instructive, rigorous and clear.

Moreover, considering the tip in the simulations is an excellent idea at the first glance. We followed that idea and performed field simulations with a gold tip at the center of the cavities. However, we quickly realized that the agreement between the new simulations with tip and the experiment becomes even worse (see Figure R1). This is caused by the movement of the tip during the measurements. Therefore, it would be necessary to move the tip in the simulations as well and obtaining field simulations for each position of the tip. This approach is not feasible.

Figure R1: Comparison field simulations with tip and without tip.

When measuring polaritons with *s*-SNOM, two different launching mechanisms can be distinguished. First, polaritons are launched by an edge and then detected by the moving *s*-SNOM tip. Second, the tip itself launches polaritons which are reflected at sample edges and boundaries and then detected by the tip. Here, we attribute the measured polaritons to edge-launched polaritons which makes including the tip in the simulations not necessary. As visible

in the propagating polaritons launched by a crystalline IST edge in Figure 2 in the main manuscript, only one oscillation caused by edge-launched polaritons appears. A second superimposed oscillation which might be caused by tip-launched polaritons is not observed. The applied fitting function only contains a single component of the polariton wavevector. If we would measure tip- and edge-launched polaritons, this would be clearly visible within the extracted line profiles.

In particular, we would like to highlight the reason: The accessible polariton wavevector induced by the tip with a radius of 30 nm displays a narrow peak at $3.4 \times 10^5 \text{ cm}^{-1}$ (Fei et al. *Nano Lett.* 11, 4701–4705, 2011) which is significantly higher than the measured polariton wavevector on SiC between $(5 - 15) \times 10^4 \text{ cm}^{-1}$. In contrast, the edges yield a broad spectrum of wavevectors with much lower accessible values. This has also been demonstrated in literature where observed polaritons can be perfectly reproduced by only assuming an edge-launched component (Huber et al. *Applied Physics Letters*, 87, 081103, 2005; Huber et al. *Journal of Microscopy* 229, 389-395, 2008). This behavior changes for polaritons on 2d materials such as hBN where the strong polariton confinement due to small flake thicknesses lead to large polariton wavevectors. Here, the tip is now able to launch polaritons which superimpose with edge launched polaritons (see Dai et al. *Science* 343, 1125-1129, 2014). Therefore, we agree with the reviewer that in the case of 2d materials the influence of the tip has to be considered, but in our case we can neglect the tip in the simulations.

Hence, polaritons on the polar crystal SiC are mainly excited by edge-launching and not by tip-launching. Thus, we did not include the tip in the simulations.

Actions taken:

- We clarified in the Methods section why we only assume edge-launched polaritons:

The s-SNOM tip is not considered in the simulations because here the dominating launching mechanism relies in edge-launching. The accessible polariton wavevectors induced by the tip are significantly larger than the measured polariton wavevectors on SiC.^[65] In contrast, the edges yield a broad spectrum of wavevectors with much lower accessible values. This has also been demonstrated in literature where observed polaritons can be perfectly reproduced by only assuming an edge-launched component.^[66,67]

[65] Z. Fei, G. O. Andreev, W. Bao, L. M. Zhang, A. S. McLeod, C. Wang, M. K. Stewart, Z. Zhao, G. Dominguez, M. Thiemens, M. M. Fogler, M. J. Tauber, A. H. Castro-Neto, C. N. Lau, F. Keilmann, D. N. Basov, *Nano Letters* **2011**, 11, 4701.

[66] A. Huber, N. Ocelic, R. Hillenbrand, *Journal of Microscopy* **2008**, 229, 389.

[67] A. Huber, N. Ocelic, D. Kazantsev, R. Hillenbrand, *Appl. Phys. Lett.* **2005**, 87, 81103

I have some concerns regarding the novelty of the work. First, there have been previous reports on reconfigurability using IST, many of them by some coauthors of the present work, so the technique/materials are not really novel. Second, as shown by the authors, the

performance metrics of the resonators are comparable to those in found in the existing literature, and some of these prior works also offer tuning capabilities, at times even in operando. Third, it seems that further refinement of the technique is necessary: reamorphization, sharpness of the edges, optimizing the layerstack and the laser induced phase-change, etc. While I acknowledge that there is potential in the concept, such potential appears to be underexplored and underexploited in the current manuscript. My concern is that modifying the confinement, fabricating squares and triangles, and simulating an hBN layer on top of some circles drawn in IST may not demonstrate the full potential of the technique and therefore may be insufficient for publication in Nature Communications. However, I would be happy to review a revised version of the manuscript if these aspects are addressed.

Our response:

1. The reviewer is correct that the used material IST is not new and has been investigated in our group extensively for metasurfaces and nanoantennas. However, the combination of IST with polariton resonators is novel and has not been demonstrated yet. Via direct laser writing, complex resonator shapes can be written and reconfigured in order to modify the observable polariton modes within the structures. Recently, the group of Peining Li also applied IST in combination with 2d anisotropic crystals such as hBN and α -MoO₃ to investigate polariton modes (Lu et al. Nanophotonics 13, 6, 2024 (ref 58 in manuscript)). Therefore, our work boosts the research area of reconfigurable polariton optics with the plasmonic PCM IST.
2. The limited performance of our resonators is only caused by the polar crystal SiC. Low-loss 2d materials such as (isotopically pure) hBN and α -MoO₃ would lead to much larger polariton propagation lengths and quality factors. However, this is a topic of a future publication where we will explore the full potential of IST combined with 2d materials.
3. The reviewer is correct that further refinements of the technique would improve the results and the influence of the edge sharpness and defects induced during reamorphization is an interesting research area. However, the induced defects and irregularities caused by the edge sharpness are much smaller than the polariton wavelength and therefore should only play a minor role. For ultra confined resonators with much smaller cavity volumes, this might change and further refinements of the methods will be required. However, the focus of this paper relies in programming and reconfiguring polariton resonator shapes directly within the plasmonic PCM IST and hence opens up a new research direction in reconfigurable polariton optics without cumbersome fabrication methods.

Actions taken:

- We highlight the novelty of our work in the discussion section:

Confined SPhP resonators are directly written and reconfigured inside IST to tune the mode confinement. The vast flexibility of our concept is shown by programming square and triangular resonators without cumbersome fabrication techniques such as electron beam lithography or focused ion beam milling. [...] Hence, the ability of programming confined resonators within the plasmonic PCM IST boosts the research area of reconfigurable polariton optics and the performance is only limited by the applied polariton-hosting material.

Minor remark: the references to the papers to twisted polaritonic materials should be revised. Refs 20, 21 are not really about doping.

Our response:

We revised the references accordingly.

Actions taken:

- We replaced the term 'doping' by intercalation.
- We added more references regarding twisted polaritonic materials:

[22] J. Duan, N. Capote-Robayna, J. Taboada-Gutiérrez, G. Álvarez-Pérez, I. Prieto, J. Martín-Sánchez, A. Y. Nikitin, P. Alonso-González, *Nano Letters* **2020**, *20*, 5323

[23] G. Hu, Q. Ou, G. Si, Y. Wu, J. Wu, Z. Dai, A. Krasnok, Y. Mazor, Q. Zhang, Q. Bao, C.-W. Qiu, A. Alù, *Nature* **2020**, *582*, 209.

[24] J. Duan, N. Capote-Robayna, J. Taboada-Gutiérrez, G. Álvarez-Pérez, I. Prieto, J. Martín-Sánchez, A. Y. Nikitin, P. Alonso-González, *Nano Letters* **2020**, *20*, 5323.